# Simplicilones A and B Isolated from the Endophytic Fungus *Simplicillium subtropicum* SPC3

**DOI:** 10.3390/antibiotics9110753

**Published:** 2020-10-29

**Authors:** Elodie Gisèle M. Anoumedem, Bel Youssouf G. Mountessou, Simeon F. Kouam, Abolfazl Narmani, Frank Surup

**Affiliations:** 1Department of Chemistry, Higher Teacher Training College, University of Yaoundé I, Yaoundé P.O. Box 47, Cameroon; gisele.elodie@yahoo.com (E.G.M.A.); mountessou@yahoo.com (B.Y.G.M.); 2Microbial Drugs Department, Helmholtz-Centre for Infection Research (HZI), Inhoffenstr. 7, 38124 Braunschweig, Germany; abolfazl.narmani2@gmail.com; 3Department of Plant Protection, Faculty of Agriculture, University of Tabriz, Tabriz 51666, Iran; 4German Centre for Infection Research (DZIF), Partner Site Hannover-Braunschweig, 38124 Braunschweig, Germany

**Keywords:** *Duguetia staudtii*, *Simplicillium*, simplicilones A and B, endophytic fungus

## Abstract

Two new tetracyclic polyketides with a spirocenter, simplicilones A (**1**) and B (**2**) were isolated from the broth-culture of the endophytic fungus *Simplicillium*
*subtropicum* (SPC3) in the course of our screening for new bioactive secondary metabolites. This endophytoic fungus is naturally harboured in the fresh bark of the Cameroonian medicinal plant *Duguetia staudtii* (Engl. and Diels) Chatrou. The planar structures of the simplicilones were elucidated by MS and 1D as well as 2D NMR spectroscopic techniques. The relative configuration was assigned by NOESY experiments in conjunction with coupling constants; subsequently, the absolute configurations were assigned by the modified Mosher’s method. The compounds showed weak cytotoxic effects against the cell line KB3.1 (in vitro cytotoxicity (IC_50_) = 25 µg/mL for **1**, 29 µg/mL for **2**), but were inactive against the tested Gram-positive and Gram-negative bacteria as well as fungi.

## 1. Introduction

Natural products continue to be the most promising source for new chemical entities, especially in the field of antibiotics and anticancer agents. Nearly two thirds of anti-cancer agents emanated from natural products in the period 1981 to the present day [1]. However, high rediscovery rates hamper the isolation of novel secondary metabolites, which might be suited as lead structures in drug development. One solution to this problem is the use of so far under-investigated organisms [2]. Plant-associated microorganisms, particularly endophytic fungi, which live inside healthy plants without evidently causing adversary effects to the host [3], constitute a relatively untapped source of secondary metabolites due to the high diversity of species [4].

As part of our ongoing screening program for bioactive secondary metabolites, endophytic fungi from grape vide and ash trees yielded the isolation of antibacterial and cytotoxic metabolites with new chemical structures [5,6,7]. Expanding this study to endophytes from African medicinal plants [8,9], strain SPC3 was isolated from a fresh bark of *Duguetia staudtii* (Engl. and Diels) Chatrou (formerly known as *Pachypodanthium staudtii* Engl. and Diels). This plant of the Annonaceae family is a large-bole tree measuring up to 40 m in height, with a straight cylindrical diameter of ca. 70 cm and long narrow leaves. It is widely distributed throughout the West and Central African regions [10], ranging from Sierra Leone to Zaire and Cameroon in the dense evergreen forest [11]. In folk medicine, various parts of this plant are used to treat several human ailments [12,13]. Previous chemical studies reported various compounds such as lignans, alkaloids, styrens, triterpenes and flavonoids [10,11,12,13,14].

From the fresh apparently healthy bark of *D. staudtii*, four endophytic fungi strains were isolated, among which was a *Simplicillium* sp. strain. The genus *Simplicillium* belongs to the Ascomycota and they have a wide host range and an extensive distribution. Since some of them play an important role in biological control [15], we investigated the strain SPC3 for its secondary metabolite production. We herein report the isolation and structural elucidation of simplicilones A and B, two secondary metabolites produced by *S. subtropicum* isolated from the Cameroonian medicinal plant *D. staudtii* (Annonaceae).

## 2. Results

A large culture of the broth-culture of the plant-derived endophytic fungus SPC3, which was identified as *S. subtropicum* (Appendix A, Appendix A), was conducted in yeast extract-malt extract-glucose (YMG) medium and a prepared crude extract was further purified by preparative HPLC, to afford two previously undescribed secondary metabolites, named simplicilones A (**1**) and B (**2**), along with a known sterol named ergosterol (Figure 1).

### 2.1. Structure Elucidation of Simplicilone A (***1***)

Simplicilone A (**1**) was obtained as a colourless oil with the molecular formula C_24_H_35_NO_5_, as deduced from the (+)-high resolution electrospray mass spectrometry (HR-ESI-MS) data (Appendix A), which showed the pseudo molecular ion peak [M + H]^+^ at *m/z* 418.2589 (calcd. 418.2593). The ^1^H NMR spectrum (Appendix A) showed signals of an olefinic proton (δ_H_ 5.72), an oxymethine proton (δ_H_ 4.04) and five methyl protons including protons of a heteroatom-linked methyl (δ_H_ 3.19). Its broad band (BB) ^13^C NMR spectrum (Appendix A) displayed signals for twenty-four carbon atoms, which were sorted by the HSQC experiment (Appendix A) into six methyl, three methylene, nine methine and six quaternary carbons including three carbonyl signals at δ_C_ 205.7, 201.6 and 172.7 (Table 1). The ^13^C NMR of **1** also showed signals at δ_C_ 132.9 and 134.0 ppm, which were assigned to the two olefinic carbons on the fused bicyclic unit. The first fused ring system (A-B), named 4,5-disubstituted-3,7-dimethylbicyclo[4.4.0]dec-2-ene, was clearly supported by the HMBC cross peaks (see Appendix A) observed between the olefinic proton signals at δ_H_ 5.72 (H-9) with the carbon signals at δ_C_ 59.2 (C-11), 37.8 (C-3) and 31.4 (C-7), and also between the methyl protons at δ_H_ 1.84 (H-15) with the carbon signals at δ_C_ 132.9 (C-9) and 134.0 (C-10) (Figure 2). The later HMBC correlations gave clear evidence for the attachment of the methyl group (δ_H_ 1.84) at the C-10 position of the fused ring unit (A-B). This ring was also supported by multiple COSY correlations as shown in Figure 2. In addition, the methyl group at δ_H_ 0.72 (H-14) was located at C-4 of the (A-B) ring, as illustrated by cross peak correlations observed between the methyl proton signals at δ_H_ 0.72 with the carbon signals at δ_C_ 39.4 (C-4), 37.8 (C-3) and 30.1 (C-5). The five-membered heterocyclic spiro ring unit (C-D), named 4,8,9-trisbustituted-3-methylazaspiro[4.5]deca-2,5,10-trione was deduced, and the chemical shift of C-2′ (δ_C_ 75.4) in compound **1** was comparable to that of a spiro quaternary carbon [16]. This spiro ring formed a bridge junction via C-2 (δ_C_ 43.2) and C-11 (59.2) of the fused ring (A-B) as a key ^3^*J* HMBC cross peak was observed between the proton signals at δ_H_ 2.11 (H-3) with the carbonyl carbon signal at δ_C_ 201.6 (C-1) and another one between the proton signals at δ_H_ 2.11 (H-3) with the spiro quaternary carbon signal (C-2′). Further HMBC cross peaks observed between the methyl proton signals at δ_H_ 1.37 (H-16) with the carbon signals at δ_C_ 77.2 (C-12) and 59.2 (C-11) and also, between the methyl proton signals at δ_H_ 0.89 (H-17) with the carbon signal at δ_C_ 53.7 (C-13), C-12 and C-2′ clearly indicated the points of attachment of these two methyl groups (C-16 and C-17) at C-13 and C-12 positions, respectively (Figure 2). Moreover, the deshielded methyl protons at δ_H_ 3.19 (H_3_-7′) resided next to the nitrogen atom as they showed HMBC cross peak correlations with the carbonyl at δ_C_ 172.7 (C-1′) and the carbon signal at δ_C_ 74.3 (C-4′). In addition, in the ^1^H NMR spectrum, the 1-hydroxyethyl moiety was deduced from the signals at δ_H_ 4.04 (1H, dq, 4.3, 7.0 Hz) and 1.34 (3H, s) which was further confirmed by the ^13^C NMR spectrum with resonances at δ_C_ 69.8 and 17.6, respectively. This moiety was attached at C-4′ of the C-D ring of compound **1** as illustrated by HMBC correlations (Figure 2) observed between the methyl protons at δ_H_ 4.04 (H-5′) with the carbon signals at δ_C_ 74.3 (C-4′) and 69.8 (C-5′) and also, between the methine proton at δ_H_ 3.78 (H-4′) with the carbon signals at δ_C_ 205.7 (C-3′), 75.4 (C-2′), 69.8 (C-5′) and 17.6 (C-6′).

Furthermore, in the rotating-frame nuclear Overhauser effect spectroscopy (ROESY) spectrum (see Figure 3 and Appendix A), correlations observed between the proton signals at δ_H_ 3.72 (H-2) with those at δ_H_ 1.37 (H-16) gave an indication of their cis-orientation, whereas in the ^1^H NMR spectrum, the proton signals of H-2 which concurrently coupled (*J* = 11.7 Hz) with both the methine proton signals at δ_H_ 2.30 (H-11) and 2.11 (H-3) indicated trans-orientations of H-2 with respect to both H-3 and H-11. Additionally, in the ^1^H NMR spectrum, the small value of the coupling constant (*J* = 3.6 Hz) observed between H-3 and H-8 (δ_H_ 1.99) also indicated a cis-juncture between the A and B rings. These results revealed that, in the tricyclic fused ring (A-B-C), the A ring formed a chair-conformation while the B and C rings formed boat-conformations as demonstrated by Koyama et al. [17]. It is noteworthy that the methine proton (H-4′) showed a cis-orientation with its vicinal homologue (H-5′) as evidenced by the small value of their coupling constant (*J* = 4.3 Hz), whereas in the ROESY spectrum, its correlation with the methyl protons at δ_H_ 0.89 (H-17) gave an indication of their cis-spatial orientation.

The modified Mosher’s method was used to determine the absolute stereochemistry at C-5′ in compound **1** [18]. (*R*)- and (*S*)-α-methoxy-α-trifluoromethylphenylacetyl (MTPA) esters of simplicilone A (**1**) were prepared in pyridine-*d*_5_ in situ by the treatment of **1** with (*S*)- and (*R*)-α-methoxy-α-trifluoromethylphenylacetyl (MTPA) chloride, respectively. Significant ∆δ values (∆δ = δ_S-MTPA-ester_ − δ_R-MTPA-ester_) were observed for the protons near the chiral centre C-5′ as shown in Figure 4. Thus, the absolute configuration at C-5′ was determined as *(R)*. This configuration was used as a starting point and consequently, the other configurations of stereogenic centres in compound **1** were determined (Figure 3) by combination of coupling constants and significant ROESY correlations. Coupling constants of *J*
_H 4′,H5′_ = 4.3 Hz, *J*
_H 4′,C5′_ = 2.5 Hz, *J*
_H 4′,C6′_ = 1.9 Hz indicated a gauche conformation between 4′-H and 5′-H as well as CH_3_-6′ and an anti-periplanar conformation between 4′-H and 5′-OH (Figure 5). The small coupling constants *J*
_H 5′,C4′_ = 2.5 Hz and *J*
_H 5′,C3′_ = 2.2 Hz indicate an anti-peri-planar conformation of 5′–H and the nitrogen atom, but a gauge conformation between 5′-H and C-3′. Taken together, this information supports a 4′*S*,5′*R* configuration. The coupling constants of *J*_H 13,C1′_ = 3.3 Hz and *J*_H 13,C3′_ = 7.5 Hz, together with the observed ROESY correlation between 17-H_3_ and 4-H, allowed to span the stereochemical information via the spiro centre to the remaining stereocenters. Thus, the structure of compound **1** was assigned to simplicilone A (Figure 1).

### 2.2. Structure Elucidation of Simplicilone B (***2***)

Compound **2**, named simplicilone B (Figure 1), was obtained as a colourless oil. Its molecular formula was determined as C_24_H_35_NO_6_ from the pseudo molecular ion peak [M + H]^+^ at *m/z* 434.2542 (calculated for C_24_H_35_NO_6_: 434.2542), obtained by (+)-HR-ESI-MS (see Appendix A) and is consistent with eight double bond equivalents. Its ^1^H, ^13^C, COSY, HSQC and HMBC NMR spectra (see Appendix A) exhibited similarities to those of compound **1**. Thus, it was obvious that the structure of compound **2** could be deduced by careful comparison of its data to those of compound **1**. Its molecular ion as shown by HR-ESI-MS differed by 16 amu from that of compound **1**, suggesting compound **2** to have an additional oxygen atom. This information was further supported by the ^13^C NMR spectrum which showed four carbonyl carbon signals (instead of three) at δ_C_ 221.5, δ_C_ 209.3 (C-1), 191.4 (C-3′) and 163.8 (C-1′). Three of them were similar to those observed in the ^13^C NMR spectrum of compound **1** (see Table 1). Although characteristic signals due to the A-B fused ring were observed in both ^1^H and ^13^C NMR spectra, in the HMBC the two methyl signals at δ_H_ 2.07 and δ_H_ 1.18 showed strong cross peaks to the additional carbonyl signal at δ_C_ 221.5, indicating the loss of the spiro centre found in compound **1**. This was further confirmed in the HMBC spectrum where the methine proton signal of the heterocyclic ring at δ_H_ 3.18 (H-4′) showed a cross peak with the deshielded sp^3^ oxygenated carbon at δ_C_ 104.9, instead of δ_C_ 75.4 for the spiro quaternary carbon (C-2′) observed in compound **1**. In the ^1^H NMR and HMBC spectra (Appendix A), signals for 2-keto-1-methylpropyl group were deduced from the signal of a three-proton singlet at δ_H_ 5.76 (H-17) which showed an HMBC cross peak with the carbonyl signal at δ_C_ 221.5 (C-13); a methyl signal at δ_H_ 1.18 displayed cross peaks with the carbon signals at δ_C_ 46.3 (C-11), 47.4 (C-12) and C-13; a methine proton doublet of quadruplet at δ_H_ 2.67 (2.5, 7.0 Hz, H-12) exhibited ^3^*J* HMBC cross peaks with the carbon signals at δ_C_ 131.4 (C-10) and 28.4 (C-10). It is noteworthy that the same ROESY correlations (Appendix A) were observed for both compounds **1** and **2**. This information was obviously expected, since the two compounds were isolated from the same organism. However, the stereochemistry at C-2′ was not determined. Based on the above investigations, the structure of compound **2** was determined as shown in Figure 1.

### 2.3. Biological Activity of Simplicilones A (***1***) and B (***2***)

To evaluate the biological activity of the metabolites, **1** and **2** were tested against a selection of microorganisms (Appendix A). However, both compounds did not show any activity against the Gram-positive bacteria *Bacillus subtilis* DSM10, *Staphylococcus aureus* DSM 346, *Micrococcus luteus* DSM 1790, *Mycolicibacterium smegmatis* ATCC 700084; the Gram-negative bacteria *Chromobacterium violaceum* DSM 30191, *Escherichia coli* DSM 1116, *Pseudomonas aeruginosa* PA14 nor the fungi *Candida albicans* DSM 1665, *Schizosaccharomyces pombe* DSM70572, *Mucor hiemalis* DSM 2656, *Pichia anomala* DSM 6766, *Rhodotorula glutinis* DSM 10134 up to 67 µg/mL.

Assessing the cytotoxicity, **1** and **2** displayed cytotoxicity with in vitro cytotoxicity (IC_50_) values of 25 and 29 µg/mL, respectively, against the cervix carcinoma cell line KB3.1, but were inactive against the murine fibroblast cell line L929 (Table 2).

## 3. Discussion

The present study describes the isolation of two compounds named simplicilones A (**1**) and B (**2**), structurally related to vermisporin PF 1052 and spylidone, two compounds produced by different fungi, *Ophiobolus vermisporus* [19] and *Phoma* sp. [17,20], respectively. They are not related to secondary metabolites isolated from the genus *Simplicillium* so far [21,22,23,24,25,26]. Simplicilone compounds have a core-shell structure related to congeners produced by other fungi [19,20]. In fact, the presence of the two methyl groups in rings A and B, respectively, as compared to the related compounds described in the literature [17,19,20], might suggest different biogenetic pathways for the new isolated compounds (**1** and **2**). This study further extends the range of compounds produced by the genus *Simplicillium* and then strengthens the assumption made by Koyama et al. [17] on the fact that spylidone (with a similar core structure with **1** and **2**) was really a metabolite produced by a fungus. Additionally, the co-occurrence of ergosterol (**3**), a typical and common metabolite of fungi further confirms the fungal origin of the two isolated compounds. The HPLC analysis of the crude extract suggested the presence of several derivatives which were not isolated due to their low amount.

Regarding the biological activity, the IC_50_ values of the two compounds (**1** and **2**) on two cancer cell lines were evaluated and the result indicates weak cytotoxic effects of the two simplicilones on the epidermoid carcinoma cell lines KB3.1, although no effect was observed on the L929 cell line. The lower IC_50_ of compound **1** on the cell line KB3 as compared to that of compound **2** may presumably be due to the spiro carbon on the C-D ring, leading to an overall more rigid backbone. Furthermore, the reinvestigation of the fungus strain SPC3 still remains challenging as well as the inhibition of lipid droplet accumulation in mouse macrophages as reported by Koyama et al. [17] for spylidone.

## 4. Materials and Methods

### 4.1. General Experimental Procedures

The ^1^H and ^13^C NMR spectra were recorded on a Bruker Advance 500 MHz spectrometer. Coupling constants are given in hertz (Hz), chemical shifts in parts per million (ppm) were referenced to the solvent signals chloroform-*d* (^1^H, δ_H_ = 7.27 ppm; ^13^C, δ_C_ = 77.00 ppm) and CD_3_OD (^1^H, δ_H_ = 4.86 ppm; ^13^C, δ_C_ = 49.15 ppm). HPLC-DAD-MS analysis was performed using an amazon speed ETD ion trap mass spectrometer (Bruker Daltonics, Billerica, Massachusetts) in positive and negative ionization modes. The mass spectrometer was coupled to an Agilent 1260 series HPLC-UV system (Agilent Technologies, Santa Clara, CA, USA) (column 2.1 × 50 mm, 1.7 μm, C18 Acquity uPLC BEH (Waters, Eschborn, Germany)). Solvent A was made up of: H_2_O + 0.1% formic acid and solvent B was consisted of acetonitrile (ACN) + 0.1% formic acid. The stepwise gradient of solvent was: 5% B for 0.5 min, increasing to 100% B in 20 min, maintaining isocratic conditions at 100% B for 10 min, flow: 0.6 mL/min, UV-vis detection 200–600 nm. HR-ESI-MS spectra were recorded on a maxis ESI TOF mass spectrometer (Bruker Daltonics, Billerica, MA, USA) (scan range m/z 100–2500, rate 2 Hz, capillary voltage 4500 V, dry temperature 200 °C), coupled to an Agilent 1200 series HPLC-UV system (column 2.1 × 50 mm, 1.7 μm, C18 Acquity uPLC BEH (Waters, Milford, MA, USA), solvent A: H_2_O + 0.1% formic acid; solvent B: acetonitrile + 0.1% formic acid, gradient: 5% B for 0.5 min, increasing to 100% B in 19.5 min, maintaining 100% B for 5 min, FR = 0.6 mL/min, UV-vis detection 200–600 nm). The molecular formulas were calculated including the isotopic pattern (Smart Formula algorithm). Preparative HPLC purification was performed at room temperature on an Agilent 1100 series preparative HPLC system (ChemStation software (Rev. B.04.03 SP1); binary pump system; column: Kinetex 5u RP C18, dimensions 250 × 21.20 mm; mobile phase: acetonitrile + 0.05% trifluoroacetic acid (TFA) and water + 0.05% TFA; flow rate 20 mL/min; diode-array UV detector).

### 4.2. Isolation of Endophytic Fungus

The endophytic fungi were isolated from the fresh bark of the apparently healthy plant *Duguetia staudtii*, collected in July 2017, at the Dja rain forest in the locality of Lomié-Bertoua (GPS coordinates provided by system WGS8: Altitude 665 m; Latitude N 4°34′38″; Longitude E 13°41′04″), in the East Region of Cameroon. The botanical identification was done by Mr. Victor Nana, a botanist at the National Herbarium of Cameroon where a voucher specimen was deposited under the number 52711HNC. The general isolation procedures were carried out as previously described by Petrini [27] with slight modifications. In brief, the fresh barks (5 × 5 cm length) of *D. staudtii* were thoroughly washed with running tap water, then sterilized in 75% ethanol for 60 s, in 3% sodium hypochlorite (NaClO) for 10 min and finally in 75% ethanol for 30 s. These barks were then rinsed three times in sterile distilled water and dried on sterile tissue paper in the laminar flow hood. Small segments were transferred to the Potato Dextrose Agar (PDA) plate supplemented with 100 mg/mL penicillin and 100 µg/mL streptomycin sulphate, then incubated at 28 °C. After a week, the purification was performed using the hyphae tip technique. This step led to the purification of four endophytic fungi on which chemical pre-screenings were carried out. Thereafter, one potential endophytic fungus (SPC3) was selected for large scale fermentation.

### 4.3. Molecular Analysis, Sequencing and Phylogenetic Analysis of Endophytic Fungus

Genomic DNA was extracted from fungal colonies growing on YMG using the EZ-10 Spin Column Genomic DNA Miniprep kit (Bio Basic Canada Inc., Markham, ON, Canada) following the manufacturer’s protocol. Molecular analysis was carried out using sequence data of internal transcribed spacer (ITS) regions. Amplification was performed using an ITS1F/ITS4 primer pair [28] in a total reaction volume of 25 µL, which was composed of 10–15 ng genomic DNA, 1× PCR buffer, 200 μM of each dNTP, 1.5 mM MgCl_2_, 0.4 pM of each primer and 0.5 U Taq polymerase. The reaction was performed on a Eppendorf PCR System with cycling conditions consisting of 5 min at 96 °C for primary denaturation, followed by 35 cycles of denaturation at 94 °C for 40 s, annealing at 52 °C for 30 s, extension at 72 °C for 60 s, with a final extension at 72 °C for 7 min. 

The amplicons were Sanger sequenced in both directions using BigDye v3.1. The resulting consensus sequence files were edited using SeqMan software in the Lasergene package (DNASTAR Inc., Madison, WI, USA) and consensus sequence was compared with sequences in the GenBank using the Basic local alignment search tool (BLAST). The ITS-rDNA sequence was deposited in GenBank with the accession number MW074157.

Reference ITS-rDNA sequence data of related/representative *Simplicillium* species were obtained from GenBank. Bayesian analyses were performed in PAUP v.4.0b10 and MrBayes v3.2.2 [29]. The most suitable model of evolution was estimated by using Mrmodeltest v.2.2 [30].

### 4.4. Fermentation, Extraction and Isolation

Pieces of a well-grown agar culture of *S. subtropicum* SPC3 were inoculated in a 500 mL-Erlenmeyer flask containing 200 mL of YMG medium consisting of 1.0% malt extract, 0.4% glucose, and 0.4% yeast extract, pH 6.3 and incubated at 23 °C. After 6 days, the cultures were harvested. The supernatant was filtrated from fungal mycelia and the latter were further extracted with acetone and the combined acetone solution was concentrated under reduced pressure to yield 2 g of dark extract. Its purification was done by preparative HPLC using a gradient of 0–15% solvent B for 3 min, 15–100% B for 20 min, and 1.0% B for 10 min. The obtained fractions were combined according to concurrent HPLC-MS-UV (DAD) analyses and yielded the isolation of compounds **1** (1.4 mg; RT = 9.1 min) and **2** (1.9 mg; RT = 12.0 min).

Simplicilones A (**1**): colourless oil, UV (MeOH): λ_max_ (PDA): 224 nm; ^1^H NMR (500 MHz, CDCl_3_) and ^13^C NMR (125 MHz, CDCl_3_) are shown in Table 1; HRMS [M + H]^+^ m/z 418.2589, (calcd. for C_24_H_36_NO_5_, 418.2593).

Simplicilones B (**2**): colourless oil, UV (MeOH): λ_max_ (PDA): 225, 287 nm; ^1^H NMR (500 MHz, CDCl_3_) and ^13^C NMR (125 MHz, CDCl_3_) are shown in Table 1; HRMS [M + H]^+^ m/z 434.2542 (calcd. for C_24_H_36_NO_6_, 434.2542). 

### 4.5. Bioassay for Cytotoxic Activity

In vitro cytotoxicity (IC_50_) of the pure compounds was investigated against the established mouse fibroblast cell line L929 (DSMZ no. ACC 2) and KB3.1 using the 3-(4,5-dimethylthiazol-2-yl)-2,5-diphenyltetrazolium bromide (MTT) method in 96-well microplates for tissue cultures. The cell line L929 was cultured in Dulbecco’s modified Eagle’s medium (DMEM; Lonza, Basel, Switzerland), supplemented with 10% foetal bovine serum (FBS; Thermo Fisher Scientific, Waltham, MA, USA) and incubated under 10% CO_2_ at 37 °C for 5 days. The assay was conducted following the procedure described previously [31].

### 4.6. Bioassay for Anti-Microbial Activity

Minimum inhibitory concentrations (MICs) of 1 and 2 were determined in a standard microdilution assay using *Bacillus subtilis* DSM10, *Staphylococcus aureus* DSM 346, *Micrococcus luteus* DSM 1790, *Mycolicibacterium smegmatis* ATCC 700084, *Chromobacterium violaceum* DSM 30191, *Escherichia coli* DSM 1116, *Pseudomonas aeruginosa* PA14, *Candida albicans* DSM 1665, *Schizosaccharomyces pombe* DSM70572, *Mucor hiemalis* DSM 2656, *Pichia anomala* DSM 6766, *Rhodotorula glutinis* DSM 10134 as test organisms for evaluating antibacterial and antifungal activities. The assay was carried out in 96-well microtiter plates with U-base (TPP Techno Plastic Products AG, Trasadingen, Switzerland) providing embroyid bodies (EBS) medium (0.5% casein peptone, 0.5% glucose, 0.1% meat extract, 0.1% yeast extract, 50 mM HEPES (11.9 g/L), pH 7.0) for bacteria and YMG medium (1.0% malt extract, 0.4% glucose, 0.4% yeast extract, pH 6.3) for yeasts and filamentous fungi. First, a stock culture of each bacterium and yeast was transferred to Erlenmeyer flasks (100 mL) filled with 30 mL of the respective growth medium. Suspensions of B. subtilis and C. albicans were incubated on a rotary shaker at 30 °C for 18–24 h, while E. coli was grown at 37 °C. Subsequently, the cultures were adjusted to cell densities of 6.7 × 10^5^ cells/mL using a hemocytometer. The spore suspension of *M. hiemalis* was prepared at a concentration of 6.7 × 10^5^ conidia/mL using YMG medium. For the test, 20 μL aliquots of 1 and 2 at 1.5 mg/mL in MeOH and 20 μL of the appropriate reference drug (broad spectrum antibiotic ciprofloxacin and antifungal cycloheximide at 1.5 mg/mL) were pipetted into the first row (A) of the 96-well microtiter plate. MeOH (20 μL) served as a negative control. Using a multichannel pipette, 150 μL of the prepared inoculum comprising the test pathogen in the respective culture medium was aliquoted in all the rows. To the first row, an additional 130 μL of the pathogen–medium mixture was added and mixed by repeated pipetting, before transferring 150 μL of this mixture to the second row. A 1:1 serial dilution was conducted in the subsequent rows to generate final compound concentrations ranging from 100 to 0.78 μg/mL. An amount of 150 μL was discarded after the last row (H). Plates were incubated at 30 °C on a microplate vibrating shaker (Heidolph Titramax 1000, Schwabach, Germany) at 600 rpm for 24–48 h. The lowest concentration of the compounds preventing visible growth of the test organism was recorded as the MIC [32].

## 5. Conclusions

This study describes the isolation of two new compounds named simplicilones A (**1**) and B (**2**), and their bioactivity on the two cancer cell lines: epidermoid carcinoma KB3.1 and L929. The structures of the new simplicilones were determined based on their NMR and HRESIMS data and their absolute stereochemistries were obtained based on the modified Mosher’s method. Although these compounds showed weak activity on the two cancer cell lines, the inhibition of lipid droplet accumulation in mouse macrophages, as reported by Koyama et al. [17] for spylidone remains to be reinvestigated.

## Figures and Tables

**Figure 1 antibiotics-09-00753-f001:**
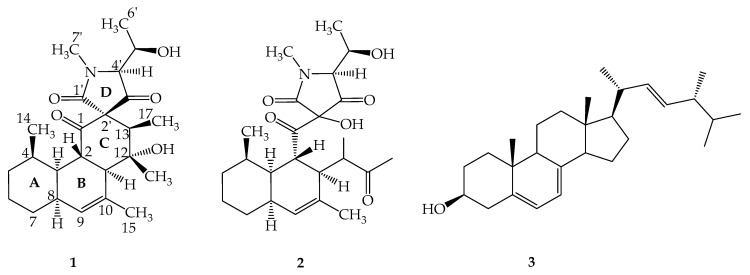
Structures of simplicilone A (**1**); simplicilones B (**2**); ergosterol (**3**).

**Figure 2 antibiotics-09-00753-f002:**
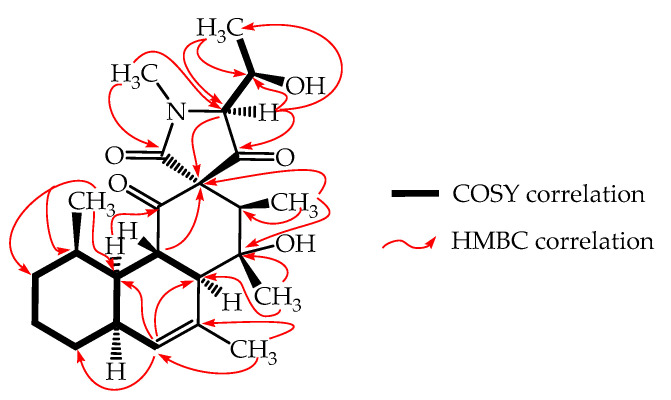
^1^H-^1^H COSY and key HMBC correlations of simplicilone A (**1**).

**Figure 3 antibiotics-09-00753-f003:**
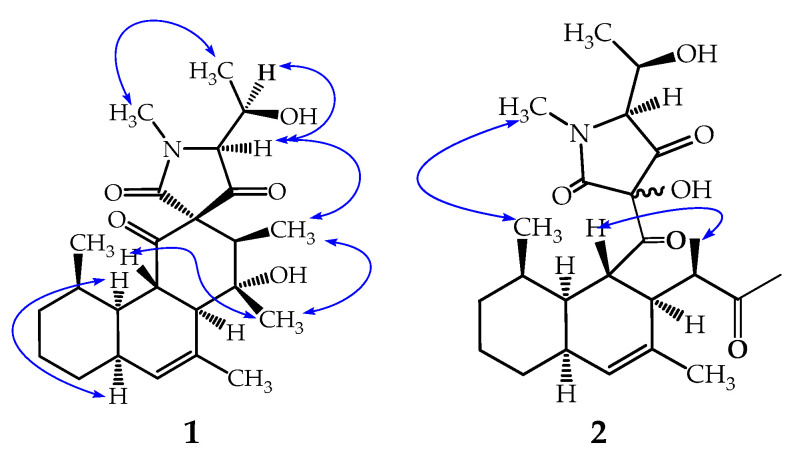
Key ROESY correlations of simplicilones A (**1**) and B (**2**).

**Figure 4 antibiotics-09-00753-f004:**
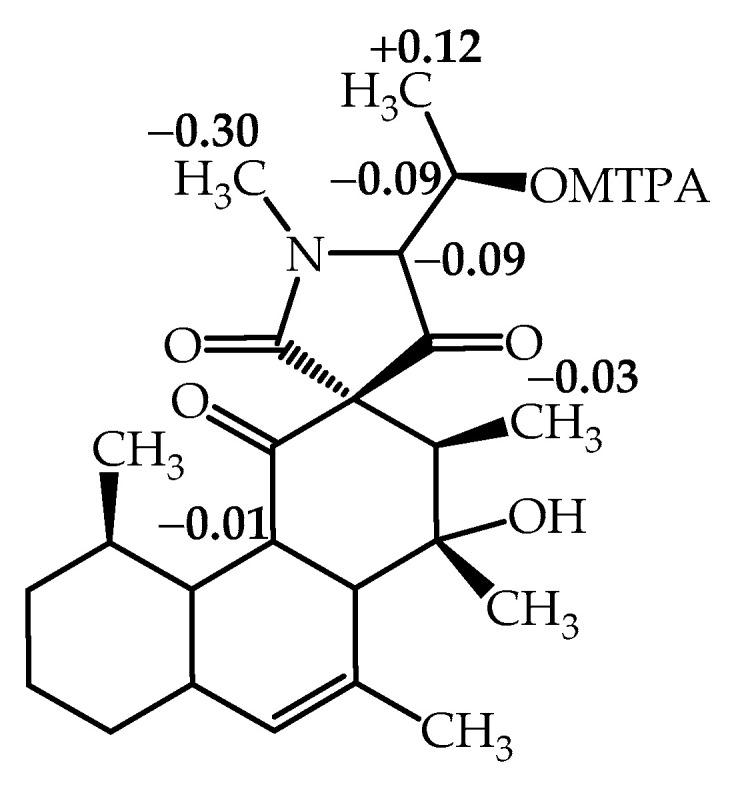
Δδ *^SR^* values (ppm) for the C-5′ α-methoxy-α-trifluoromethylphenylacetyl (MTPA) esters of compound **1**.

**Figure 5 antibiotics-09-00753-f005:**
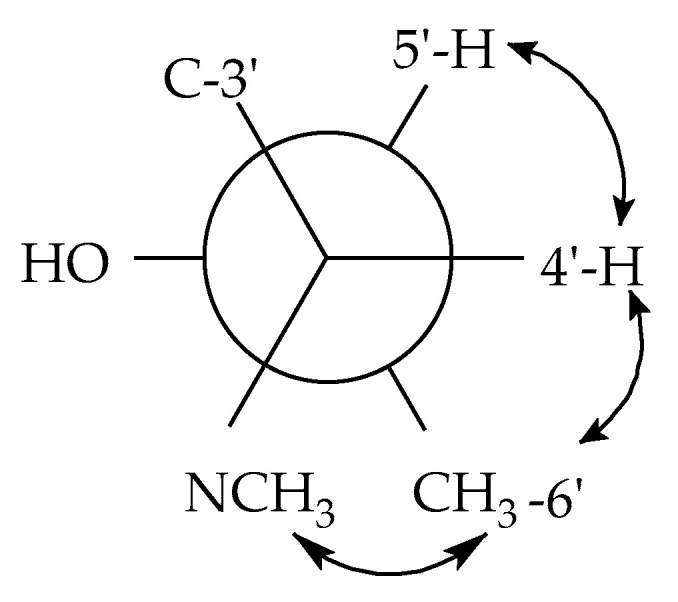
*J*-based analysis of 4′-H/5′-H bond. Arrows indicate observed ROESY correlations.

**Table 1 antibiotics-09-00753-t001:** ^1^H (500 MHz) and ^13^C (125 MHz) NMR data of compounds **1** and **2** in CD_3_OD and CDCl_3_, respectively.

Pos	1		2	
	δ_C_	δ_H_ (mult., *J* in Hz)	δ_C_	δ_H_ (mult., *J* in Hz)
1	201.7	-	209.3	-
2	43.2	3.72 (pseudo t, 11.6)	38.8	4.44 (dd, 11.6, 8.8)
3	37.9	2.11 (dt, 3.6, 11.6)	44.7	2.29 (m)
4	39.5	1.65 (m)	38.2	1.70 (m)
5	30.9	1.29 (m)	29.0	1.25 (m), 1.20 (m)
6	27.8	1.73 (m), 1.30 (m)	28.6	1.76 (m), 1.26 (m)
7	31.5	1.66 (m), 1.21 (m)	31.4	1.28 (m)
8	40.3	1.99 (m)	39.9	1.98 (m)
9	134.0	5.72 (br d, 6.7)	131.5	5.73 (br d, 6.7)
10	132.9	-	131.9	-
11	59.2	2.30 (d, 11.6)	46.4	3.22 (dd, 2.5, 11.6)
12	77.3	-	47.4	2.67 (dq, 2.5, 7.0)
13	53.7	2.51 (q, 7.0)	211.5	-
14	23.6	0.72 (d, 7.3)	21.4	0.81 (d, 7.6)
15	25.5	1.84 (s)	21.6	1.65 (s)
16	17.7	1.37 (s)	10.1	1.18 (d, 7.3)
17	11.4	0.88 (d, 7.0)	28.4	2.07 (s)
1’	172.8	-	163.8	-
2’	75.5	-	104.9	-
3’	205.8	-	191.4	-
4’	74.3	3.78 (d, 4.3)	72.8	3.18 (d, 7.6)
5’	69.8	4.04 (dq, 4.3, 7.0)	65.9	4.13 (dq, 6.4, 7.6)
6’	17.6	1.34 (d, 7.0)	18.5	1.41 (d, 6.4)
7’	31.7	3.19 (s)	45.0	2.97 (s)

**Table 2 antibiotics-09-00753-t002:** Cytotoxic activities of **1** and **2** (in vitro cytotoxicity (IC_50_) in µg/mL).

Cell Line	1	2	Reference (IC_50_)
L929 (IC_50_)	-	-	Epothilone B (0.00062)
KB3.1 (IC_50_)	25	29	Epothilone B (0.00003)

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
