# Peer review of "Simplicilones A and B Isolated from the Endophytic Fungus Simplicillium subtropicum SPC3"

_antibiotics, 2020, doi:10.3390/antibiotics9110753_

Round 1
Reviewer 1 Report
In the manuscript entitled "Simplicilones A and B isolated from the endophytic fungus Simplicillium sp. SPC3", Anoumedem et al. describe two potentially cytotoxic compounds extracted from a fungus isolated from the medicinal plant Dugettia staudii. The authors determine the structures of these compounds using a combination of mass spectrometry and NMR. The authors do a very good job of describing the structure determination of their isolated compounds, giving sufficient detail to be convincing. However, there are aspects of the reporting of data that could be improved.
The identification of the four fungi as belonging to the Simplicillium is noted in passing, but evidence is relegated to the Supplemental methods without noting this fact. Figure S1, showing a screen shot of the BLAST search is not a proper way to present these data. A proper Table should be constructed showing the scores and the E values. More importantly, a phylogenetic tree should be constructed, with a proper outgroup to show the phylogenetic relationship. This analysis should can be presented as Supplemental data, but should at least be referred to in the main text.
The conclusions regarding the cytotoxicity of these compounds is not compelling, given that only two cell lines were tested, and one of these is unaffected by either compound. The notion that these compounds would be of clinical use would be greatly strengthened if the authors could provide data on more cell lines.
The authors state in the Abstract that the compounds showed weak cytotoxicity against two cancer cell lines, which they show in Table 2. They further state that the compounds are "inactive against the tested Gram-positive and Gram-negative bacteria as well as fungi." I cannot find anywhere in the manuscript and description of such findings. Either this statement should be removed, or the data from these experiments should be presented.
Minor issues
The genus Simplicillium is spelled two different ways throughout the manuscript.
The authors need to be careful to cite all the Tables and Figures in the text. Any data, even if it is in the Supplemental Materials, need to be referred to in the text.
Author Response
Review Report #1
In the manuscript entitled "Simplicilones A and B isolated from the endophytic fungus Simplicillium sp. SPC3", Anoumedem et al. describe two potentially cytotoxic compounds extracted from a fungus isolated from the medicinal plant Dugettia staudii. The authors determine the structures of these compounds using a combination of mass spectrometry and NMR. The authors do a very good job of describing the structure determination of their isolated compounds, giving sufficient detail to be convincing.
Thank you for this assessment!
However, there are aspects of the reporting of data that could be improved. The identification of the four fungi as belonging to the Simplicillium is noted in passing, but evidence is relegated to the Supplemental methods without noting this fact. Figure S1, showing a screen shot of the BLAST search is not a proper way to present these data. A proper Table should be constructed showing the scores and the E values. More importantly, a phylogenetic tree should be constructed, with a proper outgroup to show the phylogenetic relationship. This analysis should can be presented as Supplemental data, but should at least be referred to in the main text.
We prepared a table for the results of the megablast search analysis at NCBI’s GenBank nucleotide database for strain SPC3. We also prepared informative phylogenetic tree using various reference Simplicilium species with a proper outgroup to the supporting information as suggested. We furthermore referenced to this additional data in the main text as suggested.
The conclusions regarding the cytotoxicity of these compounds is not compelling, given that only two cell lines were tested, and one of these is unaffected by either compound. The notion that these compounds would be of clinical use would be greatly strengthened if the authors could provide data on more cell lines.
We do not state that the compounds would be of clinical use for their cytotoxicity. They might be useful for a completely different scope, as the related compound spylidone is an inhibitor of lipid droplet accumulation.
The authors state in the Abstract that the compounds showed weak cytotoxicity against two cancer cell lines, which they show in Table 2. They further state that the compounds are "inactive against the tested Gram-positive and Gram-negative bacteria as well as fungi." I cannot find anywhere in the manuscript and description of such findings. Either this statement should be removed, or the data from these experiments should be presented.
We added the test results against different bacterial and fungal strains to the supporting information as suggested.
Minor issues
The genus Simplicillium is spelled two different ways throughout the manuscript.
Corrected.
The authors need to be careful to cite all the Tables and Figures in the text. Any data, even if it is in the Supplemental Materials, need to be referred to in the text.
References for all Figures and Tables were added to the manuscript as suggested.

Reviewer 2 Report
The manuscript have major drawbacks and laziness at the time of writing. An example, in line 22 the authors said that the compounds have no activity against bacteria and fungi but the methods, the strains tested, the results did not appear in any place of the manuscript. The introduction section is more an abstract section and gives no background of the work. Please modify the introduction section. It is important to emphasize the importance of this type of study, what type of similar studies have been carried out. I mean, a context. Also, the information give in the material and methods section is a bit limited and has to be improved. It is impossible to known what bioassays were carried out and the authors give some information in the conclusion section that have no correspondence with the material and methods section.
Also, what is the importance and possible application of the isolated compounds? It is not clear in any place of the manuscript. It seems only as an identification without any real application as antimicrobial for example, that is the main topic of the journal.
Other comments:
Line 32: Please avoid to use “our”, “we” and similar words.
Line 52: What known ergosterol?
Line 165: The authors give any information of the bioassays results. The authors have to include a subsection in the results section. Also, as authors give no any information of the biossays carried out, it is impossible to known what the authors did.
Line 194: The authors should reformulate the subsection order. The first section should be the isolation of the fungi.
Line 236: The authors have to include information of the method used for ITs amplication, the reagents and the equipment. Also the authors have to include how the sequencing of the amplicons were carried out and what software was used to analyze the sequence.
Line 239: Include the accession number.
Line 244: “After 6 days, the free glucose was completely depleted”. I don’t understand. Please clarify.
Line 261: Please add the information of the bioassays. It’s a bit lazy to only add a sentence with two references.
Author Response
Review Report #2
The manuscript have major drawbacks and laziness at the time of writing. An example, in line 22 the authors said that the compounds have no activity against bacteria and fungi but the methods, the strains tested, the results did not appear in any place of the manuscript.
We added the test results against different bacterial and fungal strains to the supporting information as suggested.
The introduction section is more an abstract section and gives no background of the work. Please modify the introduction section. It is important to emphasize the importance of this type of study, what type of similar studies have been carried out. I mean, a context.
We carefully revised the introduction section and added several papers to give context for our screening for bioactive metabolites.
Also, the information give in the material and methods section is a bit limited and has to be improved. It is impossible to known what bioassays were carried out and the authors give some information in the conclusion section that have no correspondence with the material and methods section.
The description of the bioassay was added to the Materials and Methods section.
Also, what is the importance and possible application of the isolated compounds? It is not clear in any place of the manuscript. It seems only as an identification without any real application as antimicrobial for example, that is the main topic of the journal.
Antibiotic is defined as á¼€ντί- "anti- „against“ und βίος bios “life” in its broad sence and is often used for bioactive secondary metabolites.
Other comments:
Line 32: Please avoid to use “our”, “we” and similar words.
We rephrased the sentence.
Line 52: What known ergosterol?
Ergosterol was isolated, too.
Line 165: The authors give any information of the bioassays results. The authors have to include a subsection in the results section. Also, as authors give no any information of the biossays carried out, it is impossible to known what the authors did.
We added the negative results in our bioassay the supporting information and the description of the bioassay to the Materials and Methods section.
Line 194: The authors should reformulate the subsection order. The first section should be the isolation of the fungi.
The subsections were organized.
Line 236: The authors have to include information of the method used for ITs amplication, the reagents and the equipment. Also the authors have to include how the sequencing of the amplicons were carried out and what software was used to analyze the sequence.
We prepared all of this information.
Line 239: Include the accession number.
We add the ITS-rDNA accession number.
Line 244: “After 6 days, the free glucose was completely depleted”. I don’t understand. Please clarify.
Rephrased.
Line 261: Please add the information of the bioassays. It’s a bit lazy to only add a sentence with two references.
We added the description of the bioassays as suggested.

Reviewer 3 Report
Although the manuscript has interesting subject, in my opinion, needs to be revised before it can be accepted for publication.
Abstract – every time abstract should contains the most important information like most important findings and results. Some values are needed. The abstract should be reorganized.
Please cite references for each information presented in the Introduction section (e.g. lines 28-29; 30-31; 39-40).
The Authors should try to make an effort to emphasize the importance of their studies.
The accepted and correct name of the studied species is Duguetia staudtii (Engl. & Diels) Chatrou. The synonym is Pachypodanthium staudtii (Engl. & Diels) Engl. & Diels.
The results and methodology of cytotoxic activity should be described.
The Authors should add details of plant material. What are the geographical coordinates of occurrence of the tested species? Which reference flora was used to identify the species?
Discussion is very important part of each manuscript published. In presented manuscript this section comprises too general explanations. Authors should discuss their results with more other scientific papers.
The conclusions should be integrated with more detailed results summarizing all the study and must reflect the innovation of this study and the perspectives.
English and style require a careful reorganization. There are a lot of language mistakes, grammatically and stylistically.
Author Response
Review Report #3
Although the manuscript has interesting subject, in my opinion, needs to be revised before it can be accepted for publication.
Abstract – every time abstract should contains the most important information like most important findings and results. Some values are needed. The abstract should be reorganized.
We revised the abstract and added the most interesting findings.
Please cite references for each information presented in the Introduction section (e.g. lines 28-29; 30-31; 39-40).
Citations were added
The Authors should try to make an effort to emphasize the importance of their studies.
We carefully revised the introduction section and added several papers to give context for our screening for bioactive metabolites.
The accepted and correct name of the studied species is Duguetia staudtii (Engl. & Diels) Chatrou. The synonym is Pachypodanthium staudtii (Engl. & Diels) Engl. & Diels.
Corrected as suggested.
The results and methodology of cytotoxic activity should be described.
We added a description
The Authors should add details of plant material. What are the geographical coordinates of occurrence of the tested species? Which reference flora was used to identify the species?
Geographical coordinated were added.
Discussion is very important part of each manuscript published. In presented manuscript this section comprises too general explanations. Authors should discuss their results with more other scientific papers.
More papers were added and discussed.
The conclusions should be integrated with more detailed results summarizing all the study and must reflect the innovation of this study and the perspectives.
We revised the conclusions section to present the key findings.
English and style require a careful reorganization. There are a lot of language mistakes, grammatically and stylistically.
We carefully proofread the manuscript.

Round 2
Reviewer 1 Report
I am satisfied with the authors' revisions.
Author Response
Thank you very much for this assessment!
Reviewer 2 Report
The authors improved the manuscript according the reviewers recommendations but there other few comments:
The authors should include the results of the antimicrobial test in the result section even if the results showed no antimicrobial activity. In the same way there is no a section for the Bioassay for cytotoxic activity and the table with the results appear in the section of structrure elucidation. Maybe the authors could include a section with the results of the cytotoxic activity and antimicrobial activity.
Another question of antimicrobial activity. Why the authors used as highest concentration 100 micrograms/mL?
Author Response
Thank you very much for this Suggestion!
We added an additional section "2.3. Biological activity of Simplicilones A (1) and B (2)" to present the biological activity more clearly.
66.7µg/ml was the highest concentration tested, since eventually all compounds will show antibiotic effect, even cooking salt.
Round 3
Reviewer 2 Report
The manuscript can be accepted now